# Instance-Specific Approximation Ratios for Correlation Clustering and Max-Cut

Sebastian Lüderssen [1]   Ioana-Oriana Bercea [2]   Stefan Neumann [1]

## Abstract

For many NP-hard optimization problems, strong theoretical inapproximability results exist. However, in practice, heuristics regularly outperform these pessimistic worst-case results on real-world datasets. Assessing the quality of these algorithms' outputs is often difficult since we lack good lower bounds on the optimal solution. In this paper, we present efficient algorithms for computing lower bounds on the optimal solutions for correlation clustering, which is a popular problem in social-network analysis. Our lower bounds allow us to provide empirical certificates that bound the solution quality of practical algorithms by obtaining *instance-specific approximation ratios*. Our main technical contribution is an algorithm that approximates an LP relaxation of a related triangle covering problem in near-linear time on sparse graphs; the algorithm is based on the multiplicative weights update framework and runs on graphs with millions of edges in a few minutes. For the concrete problem of correlation clustering, our lower bounds certify that state-of-the-art heuristics achieve almost optimal approximation ratios of 0.94 for the agreement version and 1.97 for the disagreement version (averaged over 7 real-world datasets). We also show similar results for the fundamental max-cut problem.

## 1. Introduction

The correlation clustering (CC) problem, which was introduced by Bansal et al. (2004), is an important problem in social-network analysis (Bonchi et al., 2022) and in computer vision (Keuper et al., 2015a), and it is still being actively researched in the machine learning community (Veldt et al., 2017; 2018; Veldt, 2022; García-Soriano et al., 2020; Cohen-Addad et al., 2022; 2021; Lattanzi et al., 2021).

---

[1]TU Wien, Vienna, Austria [2]KTH Royal Institute of Technology, Stockholm, Sweden. Correspondence to: Sebastian Lüderssen <sebastian.luederssen@tuwien.ac.at>.

*Proceedings of the 43rd International Conference on Machine Learning*, Seoul, South Korea. PMLR 306, 2026. Copyright 2026 by the author(s).

Despite this significant amount of research, for CC there remains a substantial theory–practice gap: many highly engineered and very efficient heuristics exist (such as the algorithm by Hausberger et al. (2023)), but they do not provide any guarantees on their approximation ratios in the worst-case. One remedy has been to provide *instance-specific approximation ratios* by computing, at runtime, a lower bound on the size of the optimal solution. Such lower bounds can subsequently be compared against a heuristic's upper bound to obtain an approximation ratio on a specific input instance (Keuper et al., 2015a; Lange et al., 2018; Veldt, 2022).

Interestingly, the best lower bounds that were obtained in this line of work are based on considering linear programming (LP) relaxations of CC or related problems (Veldt, 2022). As pointed out by Veldt (2022), the main drawback is that obtaining solutions for these LPs is highly inefficient in practice. Therefore, attention was directed towards more efficient combinatorial methods which, however, yielded worse lower bounds. This raises the following question: *Can practical algorithms compute near-optimal LP solutions in near-linear time?*

**Our contributions.** In this paper, we answer this question affirmatively and introduce novel algorithms for efficiently computing LP-based lower bounds for CC.

We obtain our lower bounds for CC by exploiting its connection to the MIN EDGE TRIANGLE COVER (MINETCOVER) problem (see Section 2 for the definition). Our main technical contribution is an algorithm that provably approximates the optimal integral LP solution for MINETCOVER in near-linear time in sparse graphs. The algorithm is based on the multiplicative weights update (MWU) framework and the fact that we need to solve a packing–covering LP, which allows us to obtain faster algorithms (Plotkin et al., 1995; Arora et al., 2012). Thus, our algorithm provides a way to obtain the highest-quality bounds identified by Veldt (2022) in an efficient and scalable way.

In practice, our algorithms allow us to compute instance-specific approximation ratios on large incomplete graphs with millions of edges in just a few minutes. In contrast, the state-of-the-art LP solver Gurobi times out after 2 hours. Further, we show that, when averaging over 7 datasets, an existing heuristic by Hausberger et al. (2023) achieves ap-

proximation ratios of 0.94 for the agreement version of CC, certifying that its solutions are just 6% worse than the optimal solution. For the disagreement version of the problem, we obtain instance-specific approximation ratios that are just 1.97 (again, averaged over 7 datasets). This is in stark contrast to worst-case hardness of approximation results (Charikar et al., 2005) showing that one cannot hope for better than an $O(\log n)$-approximation in incomplete graphs. While these are averages, some results on concrete datasets are even better: on our largest CC instance with more than 2 million edges, the approximation ratio for agreement is 0.99—indicating near-optimality—and for disagreement, it is only 1.15.

Interestingly, Fischer et al. (2025) also recently derived a similar algorithm based on the MWU framework with the same asymptotic runtime as ours. However, we show in our experiments that our algorithm is orders of magnitude faster. We achieve this by using a different strategy to update the weights. Specifically, Fischer et al. (2025) have to perform a large number of iterations since they only update a small number of weights per MWU iteration; this causes their algorithm to converge very slowly in practice. We circumvent this problem by updating a large number of weights in each iteration, allowing us to only perform a polylogarithmic number of iterations, leading to substantial improvements in practical running times. We discuss the differences between the methods in more detail at the end of Section 3.1.

We additionally introduce a simple greedy algorithm that is augmented with an initial sorting step in the preprocessing. Surprisingly, in our experiments, this algorithm yields results only slightly worse than the optimal LP solutions for CC. We further show that it outperforms the lower bound algorithm of Veldt (2022).

Finally, we use our framework to provide bounds for algorithms that compute MAX CUT solutions. Here, we again use the fact that we can bound the value of an optimal MAX CUT solution by bounding the solution for MINETCOVER. In our experiments, we find that simple combinatorial algorithms achieve approximation ratios of 0.89 (averaged over 9 datasets), beating the approximation ratio of the famous SDP-based algorithm by Goemans and Williamson (1995). This provides strong justifications for the real-world effectiveness of practical heuristics.

Our implementation is available online (Lüderssen et al., 2026).

**Related Work.** CC was introduced by Bansal et al. (2004). While the problem is NP-hard, Bansal et al. (2004) showed that in complete graphs a PTAS exists for the agreement version and that the disagreement version is APX-hard. For incomplete graphs, the agreement version also becomes APX-hard and for the disagreement version, it is

unlikely that one can do better than obtain an $O(\log n)$-approximation (Charikar et al., 2005). Prior work by Lange et al. (2018), Keuper et al. (2015a) and Veldt (2022) also obtained instance-specific lower bounds for CC and related problems, such as cluster editing (Balmaseda et al., 2024), and some of them exploited relationships to edge labeling problems based on the strong triadic closure principle (Sintos and Tsaparas, 2014). Some of these also used the same LP relaxation as we do (Ailon et al., 2008; Fischer et al., 2025; Veldt, 2022; Cao et al., 2024), but relied on slow black-box solvers to obtain optimal LP solutions.

For a detailed comparison of our MWU algorithm with other MWU approaches such as the one by Fischer et al. (2025), we refer to the end of Section 3.1.

MAX CUT is a classic NP-hard problem with a famous approximation algorithm by Goemans and Williamson (1995). Their algorithm, which heavily relies on semidefinite programming (SDP), achieves an approximation ratio of 0.878 and is optimal under the Unique Games Conjecture (Khot et al., 2007). A simple local search algorithm is known to provide a $\frac{1}{2}$-approximation. Notably, Trevisan (2012) pointed out that improving upon this approximation ratio of $\frac{1}{2}$ *without* using machinery like SDPs or spectral methods is difficult, due to a "lack of good upper bound techniques for the max cut optimum in general graphs". The results of our empirical certificates show that the real-world instances we consider are substantially easier than worst-case instances and that they allow to obtain non-trivial upper bounds based on the graphs' triangle structure. For this reason, our results do not contradict worst-case hardness results and rather show that on practical instances combinatorial heuristics can be certified to perform substantially better than worst-case guarantees would suggest.

MINETCOVER was studied from the viewpoint of attackers trying to destroy triangles in a network (Li and Yu, 2015). Li and Yu (2015) showed that a greedy algorithm computes a $1 - \frac{1}{e} \approx 0.63$ approximation due to submodularity, and they provided additional heuristic algorithms.

Instance-specific approximation ratios have also been studied for coverage problems and submodular optimization (Balkanski et al., 2021; Sharma et al., 2015; Sakaue and Ishihata, 2018; Chakrabarty and Côté, 2023; Baeza-Yates et al., 2015) using different techniques than ours.

## 2. Preliminaries

We consider graphs $G = (V, E)$ with $n$ vertices and $m$ edges. We let $\alpha$ denote the *arboricity* of $G$, which is the minimum number of forests that partition the edges of $G$. Up to a factor of 2, the arboricity is the same as the degeneracy and it is known to be small in real-world networks (Eppstein et al., 2013). We next introduce the

optimization problems we study.

**Correlation Clustering (CC).** In the CC problem (Bansal et al., 2004), the input is a signed graph $G = (V, E^+ \cup E^-)$, where edges in $E^+$ are labeled as positive and those in $E^-$ as negative. In the *disagreement* version of CC, the goal is to find an integer $k$ and a partition of the nodes $\pi : V \to \{1, \ldots, k\}$ into $k$ non-empty subsets, such as to minimize

$$\text{disagr}(\pi) = |\{uv \in E^+ \mid \pi(u) \neq \pi(v)\}|$$
$$+ |\{uv \in E^- \mid \pi(u) = \pi(v)\}|.$$

Similarly, in the *agreement* version of the problem, we wish to find an integer $k$ and a partition $\pi$ as above that maximizes

$$\text{agr}(\pi) = |\{uv \in E^+ \mid \pi(u) = \pi(v)\}|$$
$$+ |\{uv \in E^- \mid \pi(u) \neq \pi(v)\}|.$$

For CC, we recall from Bansal et al. (2004) that if $G$ contains a triangle consisting of two positive edges and one negative edge, then $\text{OPT}_{\text{CC-DIS}}(G)$ must have disagreement at least 1. We call all such triangles *bad*.

**MAX CUT.** In the MAX CUT problem, we are given a graph $G = (V, E)$ and the goal is to partition the vertices of the graph into $(S, V \setminus S)$ such that the number of cut edges is maximized, i.e., we want to maximize $|\{uv \in E \mid u \in S, v \in V \setminus S \text{ or } u \in V \setminus S, v \in S\}|$.

**MINETCOVER.** In the MIN EDGE TRIANGLE COVER problem (or MINETCOVER for short), we are given a graph $G = (V, E)$ and a set of triangles $\mathcal{T}$, where a *triangle* is a subset of edges $\{(u, v), (v, w), (u, w)\} \subseteq E$. The goal is to select a minimum number of edges whose removal makes the graph free of any triangle in $\mathcal{T}$.

**Optimal Values and Their Relationship.** Next, we formally discuss how these problems are connected with respect to their optimal objective function values.

Given a graph $G$, we denote the optimal objective function values for the disagreement and agreement version of CC, $\text{OPT}_{\text{CC-DIS}}(G)$ and $\text{OPT}_{\text{CC-AG}}(G)$, respectively, $\text{OPT}_{\text{MC}}(G)$ for MAX CUT, and $\text{OPT}_{\text{MT}}(G, \mathcal{T})$ for MINET-COVER in a graph $G$ with forbidden triangles $\mathcal{T}$. We now have the following well-known relationship between our optimization problems.

**Lemma 1.** *Let $G = (V, E^+ \cup E^-)$ be a signed graph and let $\mathcal{T}_G$ be the set of bad triangles in $G$. Then for the unsigned graph $G' = (V, E^+ \cup E^-)$, i.e., $G$ after removing edge signs, it holds that:*

1. *$\text{OPT}_{\text{CC-DIS}}(G) \geq \text{OPT}_{\text{MT}}(G', \mathcal{T}_G)$.*
2. *$\text{OPT}_{\text{CC-AG}}(G) \leq m - \text{OPT}_{\text{MT}}(G', \mathcal{T}_G)$.*

*Furthermore, let $G = (V, E)$ be an unsigned graph and let $\mathcal{T}$ consist of all triangles in $G$. Then*

3. *$\text{OPT}_{\text{MC}}(G) \leq m - \text{OPT}_{\text{MT}}(G, \mathcal{T})$.*

**Instance-specific Approximation Ratios.** Next, we describe *instance-specific approximation ratios*.

In Section 3, we show how to compute a lower bound LB for MINETCOVER, i.e., $\text{LB} \leq \text{OPT}_{\text{MT}}$. Obtaining good approximate solutions for the lower bound LB for MINET-COVER efficiently is our main challenge.

To obtain instance-specific approximation ratios for the disagreement version of CC, we proceed as follows. We use practical algorithms to compute solutions ALG for CC. Now the instance-specific approximation ratio is given by $\frac{\text{ALG}}{\text{LB}}$.

Similarly, for the maximization problems of Max-Cut and the agreement version of CC, we compute solutions ALG using practical algorithms and we compute the instance-specific approximation ratio $\frac{\text{ALG}}{m-\text{LB}}$.

We note that since LB and ALG are computed on concrete graph instances, the approximation ratios that we obtain are instance-specific and we will see in our experiments that they are substantially less pessimistic than the results obtained by theoretical worst-case approximation ratios.

Due to lack of space, we present proofs and additional experimental results in the appendix.

## 3. Lower Bounds for MINETCOVER

We present two algorithms for obtaining lower bounds on the optimal solution $\text{OPT}_{\text{MT}}$ of MINETCOVER. Through the results of Lemma 1, this enables us to also obtain bounds for CC and MAX CUT.

### 3.1. Lower Bound Through MWU Triangle Packing

Our first lower bound considers the LP primal–dual formulations of MINETCOVER. We develop an efficient approximation algorithm based on the multiplicative weight update (MWU) framework of Plotkin et al. (1995) for solving packing–covering LPs.

**Primal–Dual for MINETCOVER.** The LP relaxation of the (primal) MINETCOVER problem is defined as follows:

$$\text{minimize} \quad \sum_e y_e$$
$$\text{subject to} \quad \sum_{e \in t} y_e \geq 1 \quad \forall t \in \mathcal{T},$$
$$y_e \geq 0 \quad \forall e \in E.$$

In this formulation, we have a non-negative variable $y_e$ for each edge in the graph - in an integral solution, we would set $y_e = 1$ iff we select edge $e$ to be part of the solution to MINETCOVER. For each triangle $t \in \mathcal{T}$ we have a

constraint requiring that at least one of the edges in the triangle must be covered.

Next, we consider the dual LP, which we denote as Dual (Eq. (1)):

$$\text{maximize} \quad \sum_t x_t$$

$$\text{subject to} \quad \sum_{t:\, e \in t} x_t \leq 1 \quad \forall e \in E, \tag{1}$$

$$x_t \geq 0 \quad \forall t \in \mathcal{T}.$$

Here, we have non-negative variables $x_t$ for each triangle $t \in \mathcal{T}$. We have a constraint for each edge $e$ that ensures that the total weight of the triangles containing $e$ is at most 1.

Observe that by LP duality, any feasible solution for the Dual (Eq. (1)) is always a lower bound on the size of any integral solution for the primal MINETCOVER. Thus, in the following, our goal is to compute near-optimal feasible solutions for Dual (Eq. (1)). We note that while such LP relaxations can be solved in polynomial time, in our experiments, we will see that black-box LP solvers do not scale to instances of the size we consider.

**Efficient MWU Algorithm.** To obtain an efficient algorithm for approximately solving Dual (Eq. (1)), which is more scalable than black-box LP solvers, we present Algorithm 1, which is based on the MWU framework.

Algorithm 1 has two parts: An outer loop, denoted $\text{MWU}(\varepsilon)$, which performs a polylogarithmic number of MWU iterations, as well as an *oracle* called GREEDYBOUNDEDTRI-ANGLEPACKING. The high-level idea is as follows: we maintain a weight $w_e$ for each edge $e \in E$, corresponding to the constraints in Dual (Eq. (1)). In one iteration, the oracle is called and computes an intermediate solution that satisfies a *weighted average* of all constraints. The weights $w_e$ are then updated based on the oracle solution. Eventually, a rescaled solution is returned. We note that, even though the intermediate solutions may violate some constraints, the returned solution is feasible due to the rescaling.

Next, we provide a more detailed description of Algorithm 1. We let $A \in \{0,1\}^{|E| \times |\mathcal{T}|}$ denote the triangle-edge-incidence matrix of $G$, i.e., for each $e \in E$ and $t \in \mathcal{T}$ we set $A_{e,t} = 1$ iff $e \in t$. The algorithm computes a feasible solution $x$ for the Dual (Eq. (1)).

The procedure $\text{MWU}(\varepsilon)$ proceeds as follows. For each constraint in Dual (Eq. (1)) corresponding to an edge $e$, we define a weight $w_e$ and initialize it to 1. We also initialize $x_t \leftarrow 1$ for all triangles $t$. In each iteration $i$, $\text{MWU}(\varepsilon)$ calls the oracle and obtains an integral triangle packing $x^{(i)}$, which depends on the current edge weights $w_e$ and which satisfies $x^{(i)} \in \{0, \rho\}^{|\mathcal{T}|}$. Using this updated solution, $\text{MWU}(\varepsilon)$ keeps track of an aggregated solution $x$ as the sum

---

**Algorithm 1** MWU with Bounded Width Oracle

1: **Function** $\text{MWU}(\varepsilon)$:
2: $\quad \rho \leftarrow 3$
3: $\quad w_e \leftarrow 1 \quad \forall e \in E$
4: $\quad x_t \leftarrow 1 \quad \forall t \in \mathcal{T}$
5: $\quad$ **for** $i = 1$ **to** $2\rho \ln(m)/\varepsilon^2$ **do**
6: $\quad\quad x^{(i)} \leftarrow \text{GREEDYBOUNDEDTRIANGLEPACKING}(w, \rho)$
7: $\quad\quad x \leftarrow x + x^{(i)}$
8: $\quad\quad L_e \leftarrow (Ax)_e \qquad\qquad$ *(edge load)*
9: $\quad\quad w_e \leftarrow \left(1 + \frac{\varepsilon}{\rho}\right)^{L_e} \qquad$ *(edge weight)*
10: $\quad$ **return** $x / \max_e L_e$

11: **Function** GREEDYBOUNDEDTRIANGLEPACK-ING$(w, \rho)$:
12: $\quad x_t \leftarrow 0 \quad \forall t \in \mathcal{T}$
13: $\quad c_t \leftarrow (A^T w)_t \quad \forall t \in \mathcal{T} \qquad$ *(triangle cost)*
14: $\quad W \leftarrow \mathbf{1}^T w \qquad\qquad\qquad$ *(total weight)*
15: $\quad h_e \leftarrow 0 \quad \forall e \in E \qquad\qquad$ *(used edges)*
16: $\quad$ **Sort triangles increasingly by** $c_t$
17: $\quad$ **for triangles** $t \in \mathcal{T}$ **in sorted order do**
18: $\quad\quad$ **if** $h_e = 0$ **for all** $e \in t$ **and** $c^T x + \rho c_t \leq W$ **then**
19: $\quad\quad\quad x_t \leftarrow \rho$
20: $\quad\quad\quad h_e \leftarrow 1 \quad \forall e \in t$
21: **return** $x \qquad\qquad$ *(all entries in $\{0, \rho\}$)*

---

of all oracle solutions, and recomputes the edge loads $L_e$ and edge weights $w_e$ based on $x$. Here, for an edge $e$, its *edge load* $L_e$ is $\rho$ times the number of oracle solutions $x^{(i)}$ that contained a triangle containing $e$. After $2\rho \ln(m)/\varepsilon^2$ iterations, $\text{MWU}(\varepsilon)$ scales down the aggregated solution $x$ to ensure its feasibility and returns it.

The oracle GREEDYBOUNDEDTRIANGLEPACKING first computes a *cost* $c_t$ for each triangle $t = (u, v, w)$, defined as the sum of its three edge weights, i.e., $c_t = w_{(u,v)} + w_{(u,w)} + w_{(v,w)}$. It then sorts all triangles in increasing order of their costs and uses this ordering to construct a triangle packing (similar to Algorithm 2 below): starting with the lowest-cost triangle and iteratively adding the next cheapest triangle, which is edge-disjoint from all previously chosen ones; it only does this as long as the total cost of chosen triangles is at most $W/\rho$, or equivalently, $c^T x \leq W$. The oracle returns the incidence vector of the computed triangle packing scaled by $\rho$.

**Guarantees and Overview of the Analysis.** The guarantees of Algorithm 1 are given in the theorem below and they show that Algorithm 1 returns a solution that, up to additive error $-2$ and a factor of $1 + \varepsilon$, is at least as large as the optimal integral solution for Dual$(Eq. (1))$, which we denote $\text{OPT}_{\text{ITP}}$.

**Theorem 2.** *Algorithm 1 computes a feasible dual solution of value at least $(\text{OPT}_{\text{ITP}} - 2)/(1 + \varepsilon)$ in time*

$O\left(\alpha m \cdot \frac{\rho \log^2(n)}{\varepsilon^2}\right).$

In Appendix B, we generalize Algorithm 1 and Theorem 2 to a larger class of packing linear programs, where each variable occurs in at most $\rho$ constraints.

We prove Theorem 2 by employing the MWU framework of Plotkin et al. (1995) and by showing that our oracle has bounded width. The crucial properties that we obtain for our oracle are stated in the following lemma.

**Lemma 3.** *Let $\rho = 3$. The oracle* GREEDYBOUNDEDTRI-ANGLEPACKING *$(w, \rho)$ returns a solution $x$ satisfying*

1. $Ax \le \rho\mathbf{1}$,
2. $w^T Ax \le w^T\mathbf{1}$, *and*
3. $\mathbf{1}^T x \ge \mathsf{OPT}_{\mathsf{ITP}} - \rho + 1$.

Next, we adjust the framework of Plotkin et al. (1995) to obtain the following lemma, which allows us to show that Algorithm 1 converges after applying Lemma 3.

**Lemma 4.** *Assume access to an oracle $\mathcal{O}$ that satisfies the properties in Lemma 3 for some $\rho \ge 1$. Then the algorithm* MWU *in Algorithm 1 returns a feasible solution $x^*$ to* Dual *(Eq. (1)) with*

$$\mathbf{1}^T x^* \ge (\mathsf{OPT}_{\mathsf{ITP}} - \rho + 1)/(1 + \varepsilon)$$

*in time $O(\rho \log(n)/\varepsilon^2 \cdot T_{\mathcal{O}})$, where $T_{\mathcal{O}}$ is the time for one oracle call.*

To analyze the running time of our algorithm, we note that we can enumerate all triangles in time $O(\alpha m)$ using the algorithm by Chiba and Nishizeki (1985). Further, we note that the running time of GREEDYBOUNDEDTRIAN-GLEPACKING$(w, \rho)$ is dominated by sorting the triangles and thus takes time $O(T \log(T)) = O(\alpha m \log(n))$, where $T = |\mathcal{T}|$ is the number of triangles. Thus, overall Algorithm 1 runs in near-linear time in the number of triangles $T$; this implies that the algorithm has near-linear runtime on sparse graphs (i.e., graphs with $\alpha = \mathrm{poly}(\log n)$).

**Early Termination.** In practice, Algorithm 1 converges faster than suggested by its theoretical guarantees. Thus, we do not run precisely $2\rho \ln(m)/\varepsilon^2$ iterations, but instead compare after each iteration the packing $x/\max_e L_e$ with the feasible primal solution $\sum w_e / \min_t c_t$. We return the packing as soon as the multiplicative gap between the computed primal and dual solutions is less than $1 + \varepsilon$.

**Comparison to Existing MWU Algorithms.** Next, we briefly put the contributions of our algorithm and Theorem 2 in context with existing literature. Indeed, there exist several *theoretical* MWU algorithms that solve packing–covering LPs approximately in near-linear time. The first framework of that kind was introduced by Plotkin et al. (1995). It reduces LP solving to repeated calls to a bounded-width oracle. This approach was subsequently refined by

Garg and Könemann (2007) and Fleischer (2004), who add variable-sized weight updates to obtain width-independent algorithms. Other width-independent near-linear time approaches include Luby and Nisan (1993); Young (2001); Allen-Zhu and Orecchia (2019); Bhattacharya et al. (2023); Ju et al. (2023). However, most of these approaches require a binary search to guess the optimal solution value, which makes the running time of these algorithms infeasible in practice and, indeed, for our algorithm this is not necessary.

The work that is most related to us is by Fischer et al. (2025). They also provide a near-linear time algorithm for approximating Dual (Eq. (1)) by applying the MWU algorithm of Fleischer (2004) to the MINETCOVER problem. Interestingly, even though our Algorithm 1 and the one by Fischer et al. (2025) have the same runtime in theory (up to lower order terms), in practice our algorithm is orders of magnitude faster (see Figure 1 and Appendix C.4). The main difference is as follows: While Algorithm 1 updates the weights for an entire triangle packing in each iteration, the algorithm by Fischer et al. (2025) only updates the weight of a single triangle per iteration—resulting in a very large number of iterations until convergence. However, this difference leads to several technical challenges for obtaining our theoretical results, because analyzing our oracle that performs more-efficient batched weight updates is non-trivial. Indeed, we can only obtain our theoretical guarantees because we directly build on the framework by (Plotkin et al., 1995) and since we carefully provide the guarantees for our bounded-width oracle given in Lemma 3.

### 3.2. Lower Bound Through Triangle Packing

Next, we present another lower bound for MINETCOVER, which is based on greedy triangle packing. Concretely, consider an input graph $G$ and a set of input triangles $\mathcal{T}$. For each edge $e \in E$ we let $w_e$ denote the number of triangles containing $e$. For each triangle $t = (u, v, w) \in \mathcal{T}$, we set its *cost* $c_t \leftarrow w_{(u,v)} + w_{(v,w)} + w_{(w,u)}$ to the number of triangles that contain the edges of $t$. Now, the algorithm proceeds as follows. It sorts all triangles in increasing order of their cost $c_t$ and maintains a current set $F$ of edges, which is initially empty. Then the algorithm iterates over the triangles in increasing order of $c_t$ and, as long as there is a triangle $t$ in $\mathcal{T}$ that contains no edge from $F$, it adds all of $t$'s three edges to $F$. Eventually, it returns $|F|/3$ as a lower bound. We state the pseudocode in Algorithm 2.

We note that it is well-known that this algorithm yields a 3-approximation (since an optimal solution $\mathsf{OPT}_{\mathsf{MT}}$ for MINETCOVER must contain at least one edge for each triangle picked by Algorithm 2, but could potentially require up to three edges). Furthermore, Veldt (2022) used the same greedy algorithm *without* the additional sorting step to obtain lower bounds. In our experiments, we find that, thanks

**Algorithm 2** Lower Bound Through Triangle Packing

1: $w_e \leftarrow$ number of triangles containing $e \quad \forall e \in E$
2: $c_t \leftarrow w_{(u,v)} + w_{(v,w)} + w_{(w,u)} \quad \forall t = (u,v,w) \in \mathcal{T}$
3: Sort $\mathcal{T}$ by increasing values of $c_t$
4: $F \leftarrow \emptyset$
5: **for** $t = (u,v,w) \in \mathcal{T}$ in increasing order of $c_t$ **do**
6:     If no edge of $t$ is in $F$, add all edges from $t$ to $F$
7:     $E \leftarrow E \setminus \{(u,v),(v,w),(w,u)\}$
8: **return** $|F|/3$

*Table 1.* The datasets used in our experiments. The first 9 are unsigned and the last 7 are signed. For each graph we report its number of nodes $n$, edges $m$ and triangles $T$. For signed graphs, we report the number $T$ of bad triangles.

| instance | $n$ | $m$ | $T$ |
|---|---|---|---|
| as-caida | 26 475 | 53 381 | 36 365 |
| cond-mat | 39 577 | 175 691 | 378 059 |
| brightkite | 58 228 | 214 078 | 494 728 |
| dblp | 317 080 | 1 049 866 | 2 224 385 |
| googleplus | 211 187 | 1 165 526 | 12 918 514 |
| youtube | 495 957 | 1 936 748 | 2 443 886 |
| lastfm | 1 191 805 | 4 519 330 | 3 946 207 |
| flixster | 2 523 386 | 7 918 801 | 7 897 122 |
| flickr | 105 938 | 2 316 948 | 107 987 357 |
| bitcoinOTC | 6 005 | 21 434 | 3 875 |
| chess | 7 301 | 32 650 | 7 818 |
| wikiElec | 7 118 | 100 355 | 126 466 |
| slashdot | 82 144 | 498 532 | 64 224 |
| epinions | 131 828 | 708 507 | 397 612 |
| wikiSigned | 138 592 | 712 337 | 187 704 |
| wikiConflict | 118 100 | 2 014 053 | 2 517 267 |

to the sorting step, our algorithm achieves improvements of up to 9.5% compared to the simpler algorithm by Veldt (2022) and that on several datasets it performs almost as well as the optimal fractional LP solutions (see Appendix C.3 and Table 5).

## 4. Experiments

We compute instance-specific approximation ratios for the problems we consider on real-world datasets. For each problem, we compute lower bounds and we further run one or multiple scalable approximation algorithms to compute a solution. This allows us to evaluate the instance-specific approximation ratios our lower bounds can guarantee, and to compare them to their theoretical guarantees. Our code is available online (Lüderssen et al., 2026).

**Datasets.** We use 9 unweighted unsigned graphs for our experiments for MAX CUT and MINETCOVER and 7 signed CC instances. All instances are taken from the KONECT

network library and the NetworkRepository (Kunegis, 2013; Rossi and Ahmed, 2015). Table 1 shows key characteristics of all instances used. Additionally, we run experiments on the related CLUSTER EDITING problem using 7 further datasets (see Appendix C.3)

**Algorithms.** We implement all lower bound algorithms from Section 3 for MINETCOVER and lift them to bounds for CC and MAX CUT using Lemma 1. Below, we refer to Algorithm 2 as GREEDYPACKING. We note that all algorithms are parameter-free, except Algorithm 1, denoted MWU, which takes as input an accuracy parameter $\varepsilon > 0$. Here, we consider $\varepsilon = 0.05, 0.1, 0.5$. Furthermore, we compare our MWU algorithm to the MWU algorithm presented by Fischer et al. (2025), which updates weights only for single triangles in each iteration. We call this variant *MWU with single updates*, MWU-SU. Finally, we compare to a highly efficient bound by Balkanski et al. (2021) (we describe the details in Appendix C.2) and with the exact LP solver Gurobi (Gurobi Optimization, LLC, 2024).

To obtain solutions for the problems we study, we use a combination of simple greedy algorithms and state-of-the-art heuristics from the literature, which we describe in detail in Appendix C.1. For CC, we consider the heuristic SCMLEvo by (Hausberger et al., 2023).

**Evaluation.** In our evaluation, we follow the framework outlined in Section 2. Specifically, for each of our input graphs $G$ and for each problem, we compute an algorithmic solution $\mathrm{ALG}(G)$ and an instance-specific lower bound $\mathrm{LB}(G)$ for MINETCOVER. We then report the instance-specific approximation ratio $\frac{\mathrm{ALG}(G)}{\mathrm{LB}(G)} \in [1, \infty)$ for the disagreement version of CC and $\frac{\mathrm{ALG}(G)}{m - \mathrm{LB}(G)} \in [0, 1]$ for all other problems. Recall that a guarantee of 1 indicates an optimal solution.

We run our algorithms on a server with two 16-core AMD EPYC 9124 CPUs and 512 GB of RAM. For all algorithms, we set a timeout of 2 hours.

**Instance-specific Approximation Ratios for CC.** Table 2 shows the instance-specific approximation ratios for CC in the agreement and disagreement variants using the different lower bound methods. For the agreement version, we added a trivial bound that bounds the optimal solution by the total number of positive edges; we note that for the disagreement version, no meaningful trivial bound exists.

For the disagreement variant, our bounds can certify that, averaged over the 7 datasets, the practical SCMLEvo heuristic by Hausberger et al. (2023) generates a 1.97-approximation although it has no theoretical guarantee. This is in stark contrast to the hardness of approximation, which suggests that improving upon an approximation ratio of $O(\log n)$ cannot be done (Charikar et al., 2005).

*Table 2.* Instance-specific approximation ratios for CC per instance and lower bound method. For disagreement, lower is better; for agreement, higher is better. We write "—" to denote that a method does not finish within 2 hours. We additionally report the averaged guarantee over all instances, but only for methods without timeout, to allow for a meaningful comparison.

| | | bitcoinOTC | chess | wikiElec | slashdot | epinions | wikiSigned | wikiConflict | Average |
|---|---|---|---|---|---|---|---|---|---|
| **Disagreement** | Balkanski et al. | 2.296 | 3.326 | 1.907 | 4.988 | 2.640 | 2.229 | 1.786 | 2.739 |
| | GreedyPacking | 1.675 | 2.757 | 1.249 | 3.705 | 1.688 | 1.605 | 1.155 | 1.976 |
| | MWU-SU: $\varepsilon = 0.5$ | 1.779 | 3.045 | 1.305 | 4.024 | 1.761 | 1.701 | — | — |
| | MWU: $\varepsilon = 0.5$ | 1.724 | 2.951 | 1.337 | 3.850 | 1.732 | 1.648 | 1.183 | 2.061 |
| | MWU: $\varepsilon = 0.1$ | 1.712 | 2.833 | 1.255 | 3.806 | 1.713 | 1.632 | 1.160 | 2.016 |
| | MWU: $\varepsilon = 0.05$ | 1.708 | 2.757 | 1.237 | 3.770 | 1.700 | 1.628 | 1.146 | 1.992 |
| | Gurobi | 1.662 | 2.669 | 1.209 | 3.648 | 1.658 | 1.579 | — | — |
| **Agreement** | Balkanski et al. | 0.967 | 0.792 | 0.927 | 0.889 | 0.957 | 0.965 | 0.974 | 0.924 |
| | GreedyPacking | 0.976 | 0.807 | 0.968 | 0.898 | 0.972 | 0.976 | 0.992 | 0.941 |
| | MWU-SU: $\varepsilon = 0.5$ | 0.974 | 0.798 | 0.963 | 0.895 | 0.970 | 0.974 | — | — |
| | MWU: $\varepsilon = 0.5$ | 0.975 | 0.801 | 0.960 | 0.896 | 0.971 | 0.975 | 0.991 | 0.938 |
| | MWU: $\varepsilon = 0.1$ | 0.975 | 0.804 | 0.968 | 0.897 | 0.971 | 0.975 | 0.992 | 0.940 |
| | MWU: $\varepsilon = 0.05$ | 0.975 | 0.807 | 0.969 | 0.897 | 0.971 | 0.975 | 0.992 | 0.941 |
| | Gurobi | 0.976 | 0.810 | 0.972 | 0.898 | 0.972 | 0.976 | — | — |
| | Trivial bound | 0.942 | 0.727 | 0.859 | 0.865 | 0.933 | 0.938 | 0.942 | 0.887 |

For the agreement variant, our lower bounds certify a 0.94-approximation on average. This is a substantial improvement over the trivial bound, which only yields a 0.89-approximation on average. We note that our results here are even better than for the disagreement version, which reflects the fact that, in theory, the disagreement version has stronger hardness results than the agreement version (Bansal et al., 2004; Charikar et al., 2005).

Further, our techniques work particularly well on the large instances of our collection: On wikiConflict, which is the largest signed graph we consider, with more than 2 million edges, we obtain a 1.146 approximation guarantee in the disagreement variant and a 0.992 approximation guarantee in the agreement variant.

Comparing the different methods, we note that MWU with $\varepsilon = 0.5$ yields better lower bounds than BALKANSKI and performs similarly to MWU-SU on all datasets. While MWU-SU already times out after 2 hours on the largest instance for $\varepsilon = 0.5$, our MWU algorithm with $\varepsilon = 0.5$ finishes in less than a minute on all instances. This gives a less-than-2 approximation guarantee for the disagreement variant on average. In terms of quality, Gurobi naturally beats the MWU approaches as it computes exact LP solutions, but the runtime scales substantially worse (see also Figure 1 below), leading to timeouts on the largest instance. For CC, the GREEDYPACKING bound performs similarly to MWU for the agreement version and is marginally better for the disagreement version. On the largest instance, however, the MWU approach beats the approximation guarantee of GREEDYPACKING. We note that this is specific to CC and, in contrast, for MINETCOVER and MAX CUT, the MWU

algorithm outperforms GREEDYPACKING (see Table 3).

**Instance-specific Approximation Ratios for MINET-COVER and MAX CUT.** Table 3 shows the approximation ratios for MINETCOVER and MAX CUT.

For MINETCOVER, our lower bounds provide a 1.17-approximation guarantee, averaged over all instances. Here, our upper bound was computed using a simple greedy algorithm (see Appendix C.1) and we find it remarkable that *a simple greedy algorithm provides solutions just 17% worse than the best possible solution* (on average).

On as-caida, the GREEDYPACKING bound allows us to certify a solution quality within 2% of the optimal solution. On all other instances, the MWU bound substantially improves upon the GREEDYPACKING bound. On dblp, the MWU algorithm certifies an approximation ratio of 1.18 compared to only a 1.38 by GREEDYPACKING. The BALKANSKI bound performs worse than GREEDYPACKING on all instances except dblp, where it certifies a 1.35-approximation. The MWU-SU algorithm times out on six of the nine instances, even for a large $\varepsilon = 0.5$, thus performing worse than MWU.

For MAX CUT, our best method obtains an instance-specific approximation ratio of 0.891, averaged over the datasets. This significantly improves upon the trivial bound, which upper bounds the optimum solution by the number of edges $m$ and only achieves an average approximation ratio of 0.738. Again, the MWU methods perform best, followed by the GREEDYPACKING bounds.

**Running Time.** We report the running time for computing our lower bounds in Figure 1. Note that in the plot, the x-axis is parameterized by the number of triangles in the graph,

*Table 3.* Instance-specific approximation ratios for MINETCOVER and MAX CUT per instance and lower bound method. For MINET-COVER, lower is better; for MAX CUT, higher is better. We write "—" to denote that a method does not finish within 2 hours. We additionally report the averaged guarantee over all instances, but only for methods without timeout, to allow for a meaningful comparison.

| | | as-caida | cond-mat | brightkite | dblp | googleplus | youtube | lastfm | flixster | flickr | Average |
|---|---|---|---|---|---|---|---|---|---|---|---|
| **MINETCOVER** | Balkanski et al. | 1.845 | 1.451 | 1.943 | 1.350 | 1.994 | 1.663 | 1.662 | 2.308 | 2.076 | 1.810 |
| | GreedyPacking | 1.021 | 1.355 | 1.231 | 1.383 | 1.125 | 1.063 | 1.076 | 1.563 | 1.554 | 1.263 |
| | MWU-SU: $\varepsilon = 0.5$ | 1.104 | 1.551 | 1.326 | — | — | — | — | — | — | — |
| | MWU: $\varepsilon = 0.5$ | 1.046 | 1.453 | 1.346 | 1.423 | 1.224 | 1.116 | 1.116 | 1.662 | 1.730 | 1.346 |
| | MWU: $\varepsilon = 0.1$ | 1.034 | 1.199 | 1.165 | 1.209 | 1.118 | 1.081 | 1.083 | 1.435 | 1.538 | 1.207 |
| | MWU: $\varepsilon = 0.05$ | 1.032 | 1.165 | 1.134 | 1.178 | 1.092 | 1.062 | 1.065 | 1.397 | 1.489 | 1.179 |
| | Gurobi | 1.009 | 1.128 | — | — | — | — | — | — | — | — |
| **MAX CUT** | Balkanski et al. | 0.883 | 0.839 | 0.784 | 0.861 | 0.851 | 0.887 | 0.858 | 0.835 | 0.749 | 0.839 |
| | GreedyPacking | 0.907 | 0.858 | 0.844 | 0.855 | 0.894 | 0.914 | 0.881 | 0.898 | 0.813 | 0.874 |
| | MWU-SU: $\varepsilon = 0.5$ | 0.902 | 0.822 | 0.832 | — | — | — | — | — | — | — |
| | MWU: $\varepsilon = 0.5$ | 0.905 | 0.839 | 0.830 | 0.847 | 0.885 | 0.910 | 0.878 | 0.886 | 0.786 | 0.863 |
| | MWU: $\varepsilon = 0.1$ | 0.906 | 0.898 | 0.855 | 0.894 | 0.894 | 0.912 | 0.880 | 0.917 | 0.816 | 0.886 |
| | MWU: $\varepsilon = 0.05$ | 0.906 | 0.909 | 0.860 | 0.903 | 0.897 | 0.914 | 0.882 | 0.924 | 0.826 | 0.891 |
| | Gurobi | 0.907 | 0.921 | — | — | — | — | — | — | — | — |
| | Trivial bound | 0.855 | 0.640 | 0.699 | 0.652 | 0.801 | 0.844 | 0.820 | 0.729 | 0.606 | 0.738 |

which is the most indicative parameter for our analysis.

We find that all our methods scale linearly in the number of triangles. However, the hidden constants in the running times are considerable: The BALKANSKI bound is by far the quickest to compute, followed by the GREEDYPACKING method. We observe that the accuracy parameter $\varepsilon$ has a big impact on the running time for the MWU methods, which also corroborates the convergence bounds from Theorem 2, which quadratically depend on $\varepsilon^{-1}$. Nonetheless, these methods are highly scalable, with MWU $\varepsilon = 0.1$ running in 6 minutes on a graph with more than 2 million edges and almost 110 million triangles. The same run with $\varepsilon = 0.5$ takes only 23 seconds. Additionally, we find that MWU-SU is by far the slowest method (even slower than the exact solver Gurobi), underscoring that our MWU algorithms from Section 3.1 yield substantial improvements in practice.

**Summary.** In summary, we find that our methods yield very good instance-specific approximation ratios. Notably, for the agreement version of CC, MINETCOVER and MAX CUT, we find that practical algorithms return solutions less than 15% worse than the optimum on all but 5 datasets. For the disagreement version of CC, our instance-specific approximation ratios are substantially less pessimistic than worst-case hardness of approximation results.

Comparing the different methods, the MWU methods yield the best bounds on a majority of the instances and they scale linearly in the number of triangles. If one requires a high-quality bound even more efficiently, the GREEDYPACKING approach appears to be the most promising.

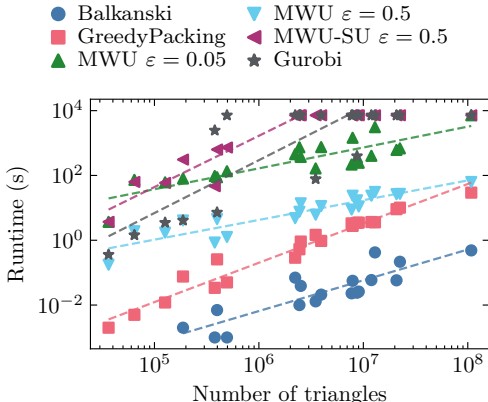

*Figure 1.* Running time comparison of the lower bound methods. All instances with at least $|\mathcal{T}| > 10^4$ are considered. We note that when a method times out, we set its running time to 2 hours; thus for Gurobi and MWU-SU the fitted lines should be even steeper.

## 5. Conclusion

We studied *instance-specific approximation ratios* for the fundamental correlation clustering (CC) and MAX CUT problems. We showed that existing practical algorithms return solutions at most 15% worse than the optimal solution on many datasets. We achieved this by exploiting a connection of these problems to triangle covering and by devising a new MWU algorithm that approximates the LP relaxation in near-linear time on sparse graphs. We also provided an improved greedy algorithm that returns solutions that can come close to the optimal fractional LP solution.

A concrete question left open by our work is whether we can obtain bounds better than the LP dual by exploiting

integrality gaps and whether some of our methods can be used to obtain better algorithmic solutions. More generally, we believe that studying instance-specific approximation ratios for other graph problems will be highly interesting. This provides a way to justify the empirical effectiveness of real-world algorithms, complementing worst-case theoretical analyses with instance-specific empirical certificates.

## Acknowledgements

We are grateful to the anonymous reviewers for their helpful suggestions that improved our paper. This research has been funded by the Vienna Science and Technology Fund (WWTF) [Grant ID: 10.47379/VRG23013]. Ioana Bercea was supported by the Swedish Research Council grant 2024-05366.

## Impact Statement

This paper presents work whose goal is to advance the field of Machine Learning. There are many potential societal consequences of our work, none of which we feel must be specifically highlighted here.

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

# A. Omitted Proofs from the Main Text

In this section, we present omitted proofs from the main text.

## A.1. Proof of Lemma 1

The results of the lemma are well-known in the literature and we present them here for the sake of completeness.

First, we prove the result for the disagreement version of CC. Consider an optimal minimum disagreement CC partition $\pi$ with disagreement $\mathsf{OPT}_{\mathsf{CC-DIS}}$. Consider all edges that violate the partition $\pi$, i.e., edges $(u, v)$ that are *either* in $E^+$ but whose endpoints satisfy $\pi(u) \neq \pi(v)$ *or* in $E^-$ that satisfy $\pi(u) = \pi(v)$. Let the set of such edges be denoted by $E'$. By definition, $\mathsf{OPT}_{\mathsf{CC-DIS}} = |E'|$. Now remove those edges from $E$ in the graph $G'$. By our previous reasoning, no bad triangles are left in $G'$ (since any bad triangle must have an edge contained in $E'$). Thus, $E'$ constitutes a valid solution to the MINETCOVER problem with $\mathcal{T} = \mathcal{T}_G$. In other words, $|E'| \geq \mathsf{OPT}_{\mathsf{MT}}$. By combining the two inequalities, the claim follows.

The result for the agreement version of CC follows by a similar argument.

Next, we prove the result for Max-Cut. Consider an optimal MAX CUT solution and let $E'$ denote the edges that are cut by it. Note that $\mathsf{OPT}_{\mathsf{MC}} = |E'|$. We claim that the edges in $E \setminus E'$ constitute a valid solution to the MINETCOVER problem. Namely, removing the edges in $E \setminus E'$ from $G$ results in a bipartite, and hence triangle-free, graph. As such, we have that $\mathsf{OPT}_{\mathsf{MT}} \leq |E \setminus E'| = m - \mathsf{OPT}_{\mathsf{MC}}$.

## A.2. Proof of Theorem 2

We now prove Theorem 2. First, we bound the performance of the oracle.

### A.2.1. PROOF OF LEMMA 3

For bounded width, note that the oracle scans triangles in nondecreasing order of cost $c_t = \sum_{e \in t} w_e$ and inserts a triangle $t$ with value $x_t = \rho$ only if the following condition is satisfied at the time of the insertion:

$$c^\top x + \rho\, c_t \leq W = w^T \mathbf{1} . \tag{2}$$

From this, we get properties (1) and (2). For near-feasibility, let $P$ denote the oracle solution and $P^*$ be the optimal integral triangle packing w.r.t cost $c$.

**Claim 5.** $|P| \geq \lfloor |P^*|/\rho \rfloor$.

The claim implies

$$\mathbf{1}^T x = \rho|P| \geq \rho \lfloor |P^*|/\rho \rfloor \geq |P^*| - \rho + 1 ,$$

and thus the lemma.

*Proof of Claim 5.* Let $T_i$ (resp. $T_i^*$) be the $i$-th lowest-cost triangle in $P$ (resp. $P^*$) with respect to the triangle cost $c$.

**Claim 6.** *For each $i$, $c_{T_i} \leq \frac{1}{3}(c_{T_{3i-2}^*} + c_{T_{3i-1}^*} + c_{T_{3i}^*})$.*

*Proof.* The statement holds for $i = 1$ as $P$ contains the triangle of minimum cost. Consider $\{T_1, \ldots, T_{i-1}\}$. At least 3 triangles $T_j^*, T_k^*, T_l^* \in \{T_1^*, \ldots, T_{3i}^*\}$ are disjoint to $\{T_1, \ldots, T_{i-1}\}$. Thus, the algorithm next picks triangle $T_i$ of cost at most

$$\begin{aligned}
c_{T_i} &\leq \min\{c_{T_j^*}, c_{T_k^*}, c_{T_l^*}\} \\
&\leq \frac{1}{3}(c_{T_j^*} + c_{T_k^*} + c_{T_l^*}) \\
&\leq \frac{1}{3}(c_{T_{3i-2}^*} + c_{T_{3i-1}^*} + c_{T_{3i}^*}),
\end{aligned}$$

which proves the claim. $\square$

We prove that the oracle does not terminate before it selects at least $\lfloor \frac{1}{3}|P^*| \rfloor$ triangles for the packing $P$. As $\rho = 3$, this proves the claim and the lemma.

First, note that any maximal triangle packing has size at least $\frac{1}{3}|P^*|$. Using Claim 6, we show that while the algorithm adds the first $\lfloor \frac{1}{3}|P^*| \rfloor$ triangles to $P$, we always have

$$
\begin{aligned}
\sum_{i=1}^{|P|} c_{T_i} &\leq \sum_{i=1}^{|P^*|/3} \frac{1}{3}\left(c_{T_{3i-2}^*} + c_{T_{3i-1}^*} + c_{T_{3i}^*}\right) \\
&= \frac{1}{3}\sum_{i=1}^{|P|^*} c_{T_i^*} \\
&\leq \frac{W}{3}.
\end{aligned}
$$

implying $c^T x \leq W$. This ensures that the weight bound of $W$ enforced by the oracle (Line 18) does not limit the size of $P$ to less than $\lfloor |P^*|/3 \rfloor$.

$\square$

### A.2.2. PROOF OF LEMMA 4

We first establish some notation. Let $w^{(i)} \in \mathbb{R}^E$ be the vector of edge weights at the beginning of iteration $i$, let $W^{(i)} := \mathbf{1}^\top w^{(i)} = \sum_{e \in E} w_e^{(i)}$ denote the total weight. Recall that $x^{(i)} \in \mathbb{R}^{\mathcal{T}}$ is the output of the oracle on weights $w^{(i)}$. We first bound the total weight $W^{(i)}$ across $\ell$ iterations as such:

**Claim 7.** *After $\ell$ iterations, it holds that:*

$$
W^{(\ell)} \leq m \cdot \exp(\ell \delta) \ ,
$$

*where $\delta = \frac{\varepsilon + \varepsilon^2}{\rho}$.*

*Proof.* In one step, we have that:

$$
w_e^{(i+1)} = w_e^{(i)} \left(1 + \frac{\varepsilon}{\rho}\right)^{s_e^{(i)}} \leq w_e^{(i)} \exp\left(\frac{\varepsilon s_e^{(i)}}{\rho}\right) \ ,
$$

where $s^{(i)} = Ax^{(i)}$ is the increase in the specific edge load due to the most recent oracle call. Because of the bounded width property (1) in Lemma 3 ($s_e^{(i)} \leq \rho$), the exponent $x = \frac{\varepsilon s_e^{(i)}}{\rho}$ satisfies $x \leq \varepsilon \leq 1$. We apply the inequality $e^x \leq 1 + x + x^2$ and get that:

$$
\begin{aligned}
W^{(i+1)} &\leq \sum_e w_e^{(i)} \left[1 + \frac{\varepsilon s_e^{(i)}}{\rho} + \left(\frac{\varepsilon s_e^{(i)}}{\rho}\right)^2\right] \\
&= W^{(i)} + \sum_e w_e^{(i)} s_e^{(i)} \left[\frac{\varepsilon}{\rho} + \frac{\varepsilon^2 s_e^{(i)}}{\rho^2}\right] \\
&\leq W^{(i)} + \sum_e w_e^{(i)} s_e^{(i)} \left[\frac{\varepsilon}{\rho} + \frac{\varepsilon^2}{\rho}\right] \ .
\end{aligned}
$$

Now we use property (2) in Lemma 3, we have that $\sum_e w_e^{(i)} s_e^{(i)} \leq W^{(i)}$ and hence, using $(1 + x) \leq e^x$:

$$
\begin{aligned}
W^{(i+1)} &\leq W^{(i)} \cdot \left(1 + \frac{\varepsilon + \varepsilon^2}{\rho}\right) \\
&\leq W^{(i)} \cdot \exp\left(\frac{\varepsilon + \varepsilon^2}{\rho}\right) = W^{(i)} \cdot \exp(\delta) \ ,
\end{aligned}
$$

After $\ell$ iterations, we have that $W^{(\ell)} \leq W^{(0)} \cdot \exp(\ell\delta)$ and we can plug in $W^{(0)} = m$ to get the claim.

$\square$

**Claim 8.** *Consider the values $L_e$ after $\ell = 2\rho\ln m/\varepsilon^2$ iterations, and let $L_{\max} = \max_e L_e$. Then:*

$$L_{\max} \leq (1 + 4\varepsilon) \cdot \ell .$$

*Proof.* After $\ell$ iterations, we have that the weight of $e$ increases, by definition, to:

$$w_e^{(\ell)} = \left(1 + \frac{\varepsilon}{\rho}\right)^{L_e} .$$

Since $w_e^{(\ell)} \leq W^{(\ell)}$, comparing with the bound in Claim 7 gives us that

$$L_e \ln\left(1 + \frac{\varepsilon}{\rho}\right) \leq \ln m + \frac{(\varepsilon + \varepsilon^2)\ell}{\rho} .$$

Using the Taylor expansion $\ln(1 + x) \geq x - x^2/2$, we get that for $\varepsilon \in (0, 1)$:

$$L_e \left(\frac{\varepsilon}{\rho} - \frac{\varepsilon^2}{2\rho^2}\right) \leq \ln m + \frac{(\varepsilon + \varepsilon^2)\ell}{\rho} .$$

Multiplying by $\rho/\varepsilon$ and using the fact that $\ell = 2\rho\ln m/\varepsilon^2$, this is equivalent to:

$$L_e\left(1 - \frac{\varepsilon}{2\rho}\right) \leq \frac{\rho\ln m}{\varepsilon} + (1 + \varepsilon)\ell = (1 + 1.5\varepsilon)\ell .$$

As $\rho \geq 1$, we have that

$$\frac{1 + 1.5\varepsilon}{1 - \frac{\varepsilon}{2\rho}} \leq \frac{1 + 1.5\varepsilon}{1 - \frac{\varepsilon}{2}} \leq 1 + 4\varepsilon .$$

This proves $L_e < (1 + 4\varepsilon)\ell$ for all edges and in particular for the edge with maximum load $L_{\max}$, which proves the claim.

$\square$

**Putting Everything Together.** To complete the proof of Lemma 4, we note that the returned solution $x^*$ is always feasible as $Ax^* = A\frac{x}{\max_e(Ax)_e} \leq \mathbf{1}$. We invoke Claim 8 by replacing $\varepsilon$ with $\varepsilon/4$ in the iteration count $\ell$ such that, after $\ell$ iterations, we get that: By the oracle property (3), the total objective value is

$$\mathbf{1}^T x \geq (\mathsf{OPT}_{\mathsf{ITP}} - 2) \cdot \ell .$$

Thus:

$$\mathbf{1}^T x^* = \frac{\mathbf{1}^T x}{L_{\max}} \geq \frac{(\mathsf{OPT}_{\mathsf{ITP}} - 2)\ell}{\ell(1 + \varepsilon)} = \frac{\mathsf{OPT}_{\mathsf{ITP}} - 2}{1 + \varepsilon} .$$

**Time Complexity.** The algorithm makes $O(\rho\ln m/\varepsilon^2)$ calls to the oracle. Hence the total running time is $O(\rho\ln m/\varepsilon^2 \cdot T_{\mathcal{O}})$. This completes the proof.

## B. Generalized Algorithm 1

We now generalize Algorithm 1 from solving triangle packing in graphs to a larger class of packing problems. Let $[n]$ be the set of integers from 1 to $n$. We consider a unit-weight packing LP with a binary matrix $A \in \{0, 1\}^{n \times m}$ of column sparsity $\rho$, i.e. each variable is contained in at most $\rho$ constraints. Our goal is to solve an LP of the form

$$\begin{aligned} \text{maximize} \quad & \mathbf{1}^T x \\ \text{subject to} \quad & Ax \leq \mathbf{1} \\ & x \geq \mathbf{0}. \end{aligned} \tag{3}$$

---

**Algorithm 3** Generalized MWU with Bounded Width Oracle

---

1: **Function** $\text{MWU}(\varepsilon, \rho)$:
2: $w_j \leftarrow 1 \quad \forall j \in [m]$
3: $x_i \leftarrow 1 \quad \forall i \in [n]$
4: **for** $k = 1$ **to** $2\rho \ln(m)/\varepsilon^2$ **do**
5: $\quad x^{(k)} \leftarrow \text{ORACLE}(w, \rho)$
6: $\quad x \leftarrow x + x^{(k)}$
7: $\quad L_j \leftarrow (Ax)_j$ *(constraint load)*
8: $\quad w_j \leftarrow \left(1 + \frac{\varepsilon}{\rho}\right)^{L_j}$ *(constraint weight)*
9: **return** $x/\max_j L_j$

10: **Function** $\text{ORACLE}(w, \rho)$:
11: $x_i \leftarrow 0 \quad \forall i \in [n]$
12: $c_i \leftarrow (A^T w)_i \quad \forall i \in [n]$ *(variable cost)*
13: $W \leftarrow \mathbf{1}^T w$ *(total weight)*
14: $h_j \leftarrow 0 \quad \forall j \in [m]$ *(used constraints)*
15: **for** $i \in [n]$ **in order sorted increasingly by** $c_i$ **do**
16: $\quad$ **if** $(A^T h)_i = 0$ **and** $c^T x + \rho c_i \leq W$ **then**
17: $\quad\quad x_i \leftarrow \rho$
18: $\quad\quad h \leftarrow h + Ae_i$
19: **return** $x$ *(all entries in $\{0, \rho\}$)*

---

Observe that if $A$ is the triangle-edge incidence matrix of a graph, this LP is equivalent to the triangle packing LP from Section 3.1.

We can give an approximate solution for this packing LP by using Algorithm 1 with an adapted oracle call. The full routine is given in Algorithm 3. The MWU routine is equivalent to the one for triangle packing, while the oracle call is generalized to compute a low-cost greedy packing of variables, i.e. no constraint depends on more than one of the selected variables. Here, the cost of a variable $x_i$ is the sum $\sum_{j:Ae_i>0} w_j$ of at most $\rho$ constraint weights, where $e_i$ denotes the $i$-th unit vector.

We obtain the following guarantee for the more general setting, where $\text{OPT}_{int}$ denotes the optimal integral solution value to the LP (Eq. (3)). For constant $\rho$, we obtain near-linear running time.

**Theorem 9.** *Algorithm 3 computes a feasible solution to (Eq. (3)) of value at least $(\text{OPT}_{int} - \rho + 1)/(1 + \varepsilon)$ in time $O(\rho n \log(n) \log(m)/\varepsilon^2)$.*

In Claim 10, we generalize Claim 5 to general values of $\rho$. Using this result, we observe that Lemma 3 also holds for general $\rho$. Furthermore, Lemma 4 already holds for general $\rho$, allowing us to prove Theorem 9 analogously to Theorem 2.

**Claim 10.** *Let $x$ denote the solution returned by $\text{ORACLE}(w, \rho)$ and $\text{OPT}_{int}$ the cost of an optimal integral LP solution w.r.t the cost vector $c = A^T w$. Then $\mathbf{1}^T x \geq \text{OPT}_{int} - \rho + 1$.*

*Proof.* The proof works analogous to the proof of Claim 5, but for general $\rho$, instead of $\rho = 3$.

Let $x^*$ be the optimal integral solution of value $\text{OPT}_{int}$, and let $P$ (resp. $P^*$) denote the set of non-zero entries in $x$ (resp. $x^*$). We say that two variables $x_a$ and $x_b$ are *disjoint* if they have disjoint support, i.e. if no constraint depends on both $x_a$ and $x_b$.

We prove the claim by showing that the oracle does not terminate before it selects at least $\lfloor |P^*|/\rho \rfloor$ variables for the packing $P$.

First, note that the number of non-zero entries in a maximal greedy packing is at least $|P| \geq |P^*|/\rho$. This holds since a packing of at most $|P^*|/\rho - 1$ variables can intersect at most $\rho(|P^*|/\rho - 1) = |P^*| - \rho$ variables of $P^*$ and thus $\rho$ variables remain disjoint out of which at least one further variable can be added to the packing. This implies that only the budget constraint $c^T x + \rho c_i \leq W$ in the oracle could prevent the size of $P$ to be at least $|P| \geq |P^*|/\rho$.

Let $(x_i)$ (resp. $(x_i^*)$) be the variables in $P$ (resp. $P^*$) sorted non-decreasingly with respect to the cost $c$, i.e. $x_i$ is the $i$-th

non-zero variable of lowest cost in $P$. Further, let $c_i$ be the cost of $x_i$ and $c_i^*$ be the cost of $x_i^*$.

**Claim 11.** *For each $i$, $c_i \leq \frac{1}{\rho} \sum_{j=0}^{\rho-1} (c_{(\rho i-j)}^*)$.*

*Proof.* The statement holds for $i = 1$ as $P$ contains the lightest variable. Consider the $i-1$ lightest variables $\{x_1, \ldots, x_{i-1}\}$ in $P$: At least $\rho$ variables $x_{j_1}^*, \ldots, x_{j_\rho}^* \in \{x_1^*, \ldots, x_{(\rho i)}^*\}$ in $P^*$ are disjoint from $\{x_1, \ldots, x_{i-1}\}$. Thus, the algorithm next picks variable $x_i$ of cost at most

$$c_i \leq \min_{k \in [\rho]}\{c_{j_k}\} \leq \frac{1}{\rho} \sum_{k=1}^{\rho} c_{j_k}^* \leq \frac{1}{\rho} \sum_{j=0}^{\rho-1} c_{(\rho i-j)}^*,$$

which proves the claim. □

Using Claim 11, we show that during the first $\lfloor \frac{1}{\rho}|P^*| \rfloor$ iteration of the oracle, we always have

$$\begin{aligned}
\sum_{i=1}^{|P|} c_i &\leq \sum_{i=1}^{|P^*|/\rho} \frac{1}{\rho} \sum_{j=0}^{\rho-1} c_{(\rho i-j)}^* \\
&= \frac{1}{\rho} \sum_{i=1}^{|P|^*} c_i^* \\
&= \frac{1}{\rho} c^T x^* \\
&= \frac{1}{\rho} w^T A x^* \\
&\leq \frac{1}{\rho} w^T \mathbf{1} \\
&= \frac{W}{\rho},
\end{aligned}$$

implying $c^T x \leq W$. This ensures that the weight bound of $W$ enforced by the oracle does not limit the size of $P$ to less than $\lfloor |P^*|/3 \rfloor$.

□

*Proof of Theorem 9.* By Lemma 4, we have that after $\Omega(\rho \log(m)/\varepsilon^2)$ iterations,

$$\mathbf{1}^T x \geq (\mathsf{OPT}_{\text{int}} - \rho + 1)/(1 + \varepsilon).$$

Thus:

$$\mathbf{1}^T x^* = \frac{\mathbf{1}^T x}{\max_j L_j} \geq \frac{(\mathsf{OPT}_{\text{int}} - \rho + 1)\ell}{\ell(1 + \varepsilon)} = \frac{\mathsf{OPT}_{\text{int}} - \rho + 1}{1 + \varepsilon}.$$

One oracle call takes time $O(n \log n)$, which gives a total running time of $O(\rho n \log(n) \log(m)/\varepsilon^2)$.

□

## C. Additional Experimental Results

In this section, we present additional information on our experiment setup and additional experimental results.

*Table 4.* Instances for CLUSTER EDITING

| instance | $n$ | $m$ | $T$ |
|---|---|---|---|
| Bowdoin47 | 2252 | 84387 | 7721082 |
| Amherst41 | 2235 | 90954 | 9041664 |
| cond-mat | 40421 | 175693 | 3495113 |
| Rice31 | 4087 | 184828 | 22408097 |
| ca-AstroPh | 18772 | 198050 | 8695057 |
| Lehigh96 | 5075 | 198347 | 20777971 |
| brightkite | 58228 | 214078 | 11939219 |

*Table 5.* Instance-specific approximation ratios for CLUSTER EDITING. We compare our lower bounds and the ones by Balkanski et al. (2021) and by Veldt (2022) to the PIVOT algorithm upper bounds.

| | Bowdoin47 | Amherst41 | cond-mat | Rice31 | AstroPh | Lehigh96 | Brightkite | Average |
|---|---|---|---|---|---|---|---|---|
| Balkanski et al. (2021) | 2.760 | 2.707 | 2.709 | 2.854 | 2.741 | 2.948 | 2.943 | 2.809 |
| Veldt (2022) | 2.358 | 2.305 | 2.268 | 2.384 | 2.154 | 2.468 | 2.345 | 2.326 |
| GreedyPacking | 2.352 | 2.299 | 2.051 | 2.379 | 2.059 | 2.462 | 2.242 | 2.263 |
| MWU-SU: $\varepsilon = 0.5$ | — | — | — | — | — | — | — | — |
| MWU: $\varepsilon = 0.5$ | 2.360 | 2.304 | 2.134 | 2.396 | 2.111 | 2.466 | 2.297 | 2.295 |
| MWU: $\varepsilon = 0.1$ | 2.365 | 2.299 | 2.104 | 2.391 | 2.088 | 2.463 | 2.274 | 2.283 |
| MWU: $\varepsilon = 0.05$ | 2.363 | 2.298 | 2.096 | 2.388 | 2.082 | 2.463 | 2.268 | 2.280 |
| Gurobi | — | — | 2.046 | — | 2.056 | — | — | — |

## C.1. Practical Heuristics Used in Our Experiments

We now describe which heuristics we use to obtain solutions for the problems we study.

For CC we use SCMLEVO by Hausberger et al. (2023) which is a highly engineered algorithm that provides excellent results also on incomplete graphs. In preliminary experiments, we also used the algorithms by Keuper et al. (2015b) but found that their solutions do not improve upon SCMLEVO.

For MINETCOVER we use a simple greedy algorithm, denoted SETCOVERGREEDY, which works as follows: As long as there exists an uncovered triangle, we add the edge with highest $\Delta_e$-value to our cover. To make the algorithm efficient, we maintain the edges' $\Delta_e$-values in a heap and after each iteration we only update those $\Delta_e$-values that were affected by the previously picked edge.

For MAX CUT we use the standard local search algorithm, denoted LOCALSEARCH, where we initialize a random cut by assigning each node to one of the cut sides uniformly at random and then flip the assignment of single nodes until the solution can no longer be improved. We use the best result from 10 independent runs of LOCALSEARCH. Further, we use the two fast MAX CUT heuristics DUARTE (Duarte et al., 2005) and FESTA (Festa et al., 2002) which performed well in the comprehensive evaluation of Dunning et al. (2018) and which yield better solutions than the simple LOCALSEARCH; we used the implementations by Dunning et al. (2018).[1] The best of the three computed solutions is used for the results in Table 3.

## C.2. Lower Bound by Balkanski et al. (2021)

We observe that we can use ideas from Balkanski et al. (2021) to obtain a feasible fractional solution for Dual (Eq. (1)) which thus also serves as a lower bound for MINETCOVER.

**Lemma 12** (Balkanski et al. (2021)). *It holds that*

$$\sum_{t \in \mathcal{T}} \min_{e \in T} \frac{1}{\Delta_e} \leq \mathsf{OPT}_{\mathsf{MT}} ,$$

---

[1] https://github.com/MQLib/MQLib

---

**Algorithm 4** MWU with Single Triangle Updates

---

1: **Function** $\mathrm{MWU}(\varepsilon)$:
2:    $s \leftarrow 0$
3:    $w_e \leftarrow 1 \quad \forall e \in E$
4:    $x_t \leftarrow 0 \quad \forall t \in \mathcal{T}$
5:    **while** $\mathbf{1}^T w < (m(1+\varepsilon))^{1/\varepsilon}/(1+\varepsilon)$ **do**
6:      $c_t \leftarrow (A^T w)_t \quad \forall t \in \mathcal{T}$             *(triangle cost)*
7:      $i \leftarrow \arg\min_{t \in \mathcal{T}} c_t$
8:      $x_i \leftarrow x_i + 1$
9:      $L_e \leftarrow (Ax)_e \quad \forall e \in E$             *(edge load)*
10:     $w_e \leftarrow (1+\varepsilon)^{L_e} \quad \forall e \in t$          *(edge weight)*
11:     $s \leftarrow \max\{s, \mathbf{1}^T x / \max_e L_e\}$
12: **return** $s$

---

where $\Delta_e = |\{t \in \mathcal{T} \mid e \in t\}|$ *is the number of triangles that the edge $e$ covers. This bound can be computed in time* $O(\alpha m)$.

### C.3. Cluster Editing

In CLUSTER EDITING, we are given an undirected and unsigned graph $G$ and aim to add or remove the minimum number of edges s.t. $G$ is turned into a disjoint union of cliques. This problem is equivalent to CC on complete graphs as we can interpret $G$ as the positively signed edges in an instance of CC with all other edges getting a negative sign. The number of bad triangles in the CC instance corresponds to the number of open wedges in $G$.

We compare our methods to Veldt et al. (2018). In Table 4 we list the instances used for our CLUSTER EDITING experiments. Although the number of nodes and edges of these graphs is rather low, the number of bad triangles is large compared to our instances for MAX CUT and CC in Table 1, as most of the edges of the implicit complete graph are negative and thus not part of the input.

We run the classical randomized PIVOT algorithm on each instance to obtain an upper bound by using the best result of 50 runs. We present our results in Table 5.

We observe that GREEDYPACKING and MWU perform better than Veldt (2022) on most instances, in particular on cond-mat and brightkite. GREEDYPACKING slightly outperforms the MWU methods for this problem. MWU again runs way faster than MWU-SU.

### C.4. Comparison of Practical MWU algorithms

In Algorithm 4 we provide the pseudocode for the MWU algorithm using single triangle updates from (Fischer et al., 2025) via (Garg and Könemann, 2007; Fleischer, 2004).

This algorithm is simpler than Algorithm 1 as it queries just a single triangle in each MWU iteration instead of a triangle packing. Each edge weight can be increased at most $O(\log_{1+\varepsilon}(m^{1/\varepsilon})) = O(\log(m)/\varepsilon^2)$ iterations. After updating an edge weight, all incident triangle costs are updated. As this implies each triangle cost is updated at most $O(\log(m)/\varepsilon^2)$ times, the total amortized runtime to find the minimum weight triangle and thus the total runtime is bounded by $O(T \log(m)/\varepsilon^2)$.

Although the theoretical running time is asymptotically equal, including the dependency on $\varepsilon$, in practice Algorithm 4 requires significantly more iterations to converge.

**MWU Convergence.** We explore the convergence rates next. We provide a sensitivity analysis on the parameter $\varepsilon$ in the MWU-based methods (Algorithm 1) and compare its convergence to Algorithm 4. Figure 2 shows the result. We consider the best primal and dual solution computed by the MWU algorithm after each iteration and plot the multiplicative gap, that is the current approximation guarantee. The curve stops when the desired $(1+\varepsilon)$-approximation is found.

We observe that MWU-SU (Algorithm 4) requires nearly two orders of magnitude more time to obtain a 1.5-approximation guarantee than MWU.

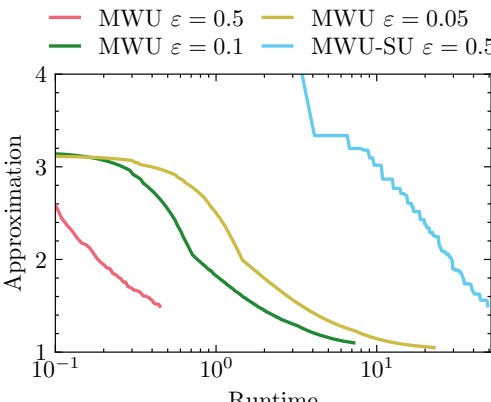

*Figure 2.* Certified approximation ratio over time for different MWU algorithms on wikiElec. MWU-SU (Algorithm 4) requires nearly two orders of magnitude more time to achieve an approximation ratio of 1.5 compared to our oracle-based MWU.

## C.5. Space Complexity

In theory, GREEDYPACKING requires linear space $O(T)$, and the MWU algorithm requires space $O(m+T)$. In practice, we ran additional experiments to evaluate our algorithms' memory usage on our 4 largest instances for Correlation Clustering (slashdot, epinions, wikiSigned, and wikiConflict) and on 5 of our largest Max-Cut instances (dblp, googleplus, youtube, lastfm, and flixster). For Correlation Clustering, we observe that on average, MWU uses less than 1.05x more memory than GREEDYPACKING. In contrast, Gurobi used 10x more space than MWU on average. For Max-Cut, MWU used 1.07x more space than GREEDYPACKING, but 13x less space than Gurobi on average.

