# OpenReview forum: "Instance-Specific Approximation Ratios for Correlation Clustering and Max-Cut"
_ICML.cc/2026/Conference — ICML 2026 regular_

### Official Review · Reviewer_vznk · 2026-03-12

**Soundness:** 3
**Presentation:** 3
**Significance:** 3
**Originality:** 3
**Overall Recommendation:** 5
**Confidence:** 2

**Summary:**

This paper is on instance specific approximation ratios for max cut and correlation clustering. They develop fast algorithms that provide lower bounds given an instance, using the relationship between min edge triangle cover problem, max cut and correlation clustering. They  prove theoretically that their lower bounds are valid, show that they are fast and confirm this on experiments.

**Compliance With Llm Reviewing Policy:**

Affirmed.

**Final Justification:**

I stand by my score.

**Key Questions For Authors:**

- what is the memory usage of your algorithm? for large scale graphs, memory is as important as computation time.

- in the intro you mention that you approximate the optimal LP solution for min edge triangle cover (which is OPT_LP), however your bound is in terms of OPT_ILP.

**Limitations:**

Yes

**Strengths And Weaknesses:**

The paper is well written. The specific ideas are not original (MWU, using min edge triangle cover, LP), but they use these ideas nicely to get the speed up they want.

---

> ### Author Rebuttal · Authors · 2026-03-31
>
> We thank the reviewer for the positive evaluation of our paper.
>
> > What is the memory usage of your algorithm?
>
> We will include both theoretical and empirical memory analyses in the final version. In theory, GreedyPacking uses space linear in the number of triangles $T$, and the MWU algorithm requires space $O(m + T)$. In practice, we ran additional experiments to evaluate our algorithms' memory usage on our 4 largest instances for Correlation Clustering (slashdot, epinions, wikiSigned, and wikiConflict) and on 5 of our largest Max-Cut instances (dblp, google+, youtube, lastfm, and flixster). For Correlation Clustering, we observe that on average, MWU uses less than 1.05x more memory than GreedyPacking. In contrast, Gurobi used 10x more space than MWU on average.
> For Max-Cut, the MWU used 1.07x more space than GreedyPacking, but 13x less space than Gurobi on average.
>
> > in the intro you mention that you approximate the optimal LP solution for min edge triangle cover (which is OPT_LP), however your bound is in terms of $OPT_{ILP}$.
>
> We thank the reviewer for pointing out this ambiguity. We will change it as follows: "Our main technical contribution is an algorithm that provably approximates the optimal *integral* LP solution for \textsc{MinETCover} in near-linear time in sparse graphs."
>
> We additionally note that, while the theoretical worst case bound is in terms of $OPT_{ILP}$, we observe that this is not an issue on real-world instances in practice, as our algorithm always certifies an $(1+\varepsilon)$-approximation to $OPT_{LB}$ using a feasbible primal/dual solution pair.

---

> > ### Author Rebuttal · Reviewer_vznk · 2026-04-02
> >
> > thanks, will keep my score.

---

### Official Review · Reviewer_k8Y3 · 2026-03-12

**Soundness:** 3
**Presentation:** 3
**Significance:** 3
**Originality:** 3
**Overall Recommendation:** 5
**Confidence:** 4

**Summary:**

This paper provides a more efficient lower/upper bound scheme for some classic clustering problems: correlation clustering (both min disagreement and max agreement objectives), max cut and min triangle deletion. They use this new scheme to obtain empirical approximation ratios for a number of large real world instances.

**Compliance With Llm Reviewing Policy:**

Affirmed.

**Final Justification:**

the extension to general sparse constraints is a broad class of packing-covering LPs.

**Key Questions For Authors:**

Can you phrase your result as something that applies to any "sparse" covering/packing problem, where the #nonzeros/row of the covering problem is small? Then the triangle cover problem is just one special case, but your efficient MWU may be more broadly useful.

**Limitations:**

yes

**Strengths And Weaknesses:**

Their approach to getting bounds on the optimal value starts with some known observations to “reduce” the task to a linear program (LP) relaxation for min triangle deletion, where we need to select edges so that all triangles in a given set T are “hit”. This is fractional covering LP for which there are “efficient” combinatorial approximation schemes using multiplicative weight updates (MWU),  see [Plotkin + 1995, survey by Arora+ ]. The authors make use of this framework but add some additional insights which lead to a practical speedup

(1) they select more triangles in the “oracle” for the dual packing problem (in the usual MWU which applies broadly to any covering/packing LP the oracle would use just one packing variable).

(2) They do not need to perform binary search on the optimal value.

(3) They stop early if the gap between feasible primal & dual solutions are small enough.

Idea #1 seems the most interesting one and this relies on the fact that their packing constraints are “3 sparse”, as seen in the proof of Lemma 3. This is a nice application of MWU that needs a slight modification of the classic framework. The new ideas/proofs are not surprising but turn out to be quite useful for their application.

They also report a lot of  experimental results on large instances with ~100k nodes. They compare known heuristics to the bounds produced by their approximate LP method (and a previous method [Balkanski] and the Gurobi solver). Their LP solver takes around 6 minutes for approx. parameter \eps=0.1 and is much faster than Gurobi. It provides empricial approximation ratios of around 2 for min disagreement, around 0.95 for max-agreement and around 0.9 for max-cut.

Minor comments:
In line 80, 2nd column: you compare your empirical finding on max-cut to various theoretical results. It is misleading to state “we bypass this theoretical limitation by..” Indeed, you provide instance-specific empirical approximation ratios. But what is the theoretical limit that was bypassed? Your method does not provide any theoretical approximation results at all. There is no reason to expect your instances to behave like worst-case instances for max-cut.

---

> ### Author Rebuttal · Authors · 2026-03-31
>
> We thank the reviewer for the positive evaluation of our paper.
>
> We are grateful for the suggestion of extending our MWU algorithm to general sparse packing--covering LPs.  We have worked through the details and agree that this is possible. We will incorporate the following extension of our theoretical work in the final version of the paper: We will show that the MWU algorithm can be directly adapted to the general setting, where the number of iterations will scale logarithmically in the number of constraints of the packing problem.  To build an oracle, we will assume that the packing LP is sparse in the sense that each variable is used on at most $\rho$ constraints.  The oracle algorithm can then be extended to the general case by iterating over packing variables sorted by their cost and collecting a set of variables with disjoint constraint sets up to a bounded total weight. For the analysis of the general oracle, we adapt Lemma 3 based on our sparsity assumption. The running time of the general algorithm will be bounded by the number of packing variables and the problem width.
>
> We will also address the reviewer's minor comment by removing the misleading sentence and simply pointing out that on the (non-worst-case) real-world instances we consider, it is considerably easier to compute good upper bounds.

---

> > ### Author Rebuttal · Reviewer_k8Y3 · 2026-04-03
> >
> > Thanks for the response. I will raise my score to accept.

---

### Official Review · Reviewer_4W9a · 2026-03-12

**Soundness:** 3
**Presentation:** 3
**Significance:** 3
**Originality:** 3
**Overall Recommendation:** 4
**Confidence:** 2

**Summary:**

This paper presents efficient algorithms for computing lower bounds on the optimal solutions for correlation clustering.They develop an algorithm that approximates an LP relaxation for a related triangle covering problem in nearly linear time on sparse graphs, utilizing the multiplicative weights update framework. Although Algorithm 1 and the one proposed by Fischer et al. (2025) theoretically have the same runtime, the algorithm presented in the paper is significantly faster in practice. The key distinction lies in that Algorithm 1 updates the weights for the entire triangle packing during each iteration, while Fischer et al.'s algorithm only refreshes the weight of a single triangle per iteration.

**Compliance With Llm Reviewing Policy:**

Affirmed.

**Final Justification:**

I will still maintain my previous score.

**Key Questions For Authors:**

I am wondering if the authors could provide more theoretical results.

**Limitations:**

Did not discuss the limitations.

I am wondering if the authors could provide more theoretical results.

**Strengths And Weaknesses:**

The experiments are well-designed and the improvement of the proposed method is apparent. The theoretical results is relatively not enough. Overall, the paper addresses an important and  relevant problem for fast algorithms of correlation clustering.

---

> ### Author Rebuttal · Authors · 2026-03-31
>
> We thank the reviewer for the positive evaluation of our paper.
>
> > I am wondering if the authors could provide more theoretical results.
>
> Based on the question of Reviewer k8Y3, we will generalize our theoretical results for the MWU algorithm to a broader class of sparse packing--covering LPs. Beyond solving the special case of Triangle Packing, we can extend the algorithm and oracle to solve any sparse packing--covering LP with small width efficiently. Specifically, the running time of this general algorithm will depend on the width of the problem and the number of constraints in the packing LP. This significantly generalizes our theoretical contributions and makes them more applicable to other problems.

---

> > ### Author Rebuttal · Reviewer_4W9a · 2026-04-02
> >
> > Thanks for the author's reply.  I will still maintain my previous score.

---

### Official Review · Reviewer_Bkut · 2026-03-13

**Soundness:** 4
**Presentation:** 4
**Significance:** 3
**Originality:** 3
**Overall Recommendation:** 5
**Confidence:** 4

**Summary:**

While theoretical inapproximability results for NP-hard optimization problems suggest limited performance guarantees for heuristics, these results often fail to predict how such methods perform on real-world instances. To bridge this gap, the authors propose studying instance-specific approximation ratios by deriving instance-wise lower bounds (for minimization problems) and comparing them to the best feasible solutions found.

The work centers on three optimization problems: Correlation Clustering, Max-Cut, and MinETCover (Min Edge Triangle Cover problem). The authors introduce two algorithms to compute lower bounds for the MinETCover problem:  a simple greedy packing algorithm, and a method based on the Multiplicative Weights Update framework (MWU). These algorithms also yield direct bounds for Correlation Clustering and Max-Cut.

The authors evaluate their approach on 16 real-world datasets, comparing their lower bounds to the best solutions obtained using state-of-the-art heuristics. Their results reveal two key findings: Their bounds significantly outperform existing theoretical guarantees from the literature. ​​Existing heuristics often produce solutions very close to optimal (typically within 15%), even for large-scale graphs.

**Compliance With Llm Reviewing Policy:**

Affirmed.

**Final Justification:**

The paper addresses a fundamental problem in AI and introduces novel bounds that outperform those in the existing literature, using innovative analytical techniques. Its contribution can be further strengthened by an extension to sparse packing-covering linear programs thanks to another reviewer remark.

The strengths of the paper clearly outweigh its weaknesses. Therefore, I am increasing the originality score to Good and my overall estimation to Accept.

**Key Questions For Authors:**

One aspect that would further clarify the paper’s contributions is the originality of the analytical approach for your MWU-based algorithm. While the algorithm builds on the framework introduced by Fischer et al. (2025), your work updates the weights of all triangles simultaneously, leading to improved lower bounds and computational efficiency. Could you elaborate on the novelty in the analysis itself?

**Limitations:**

yes

**Strengths And Weaknesses:**

Strengths

Significance and Contributions.
- The paper addresses Correlation Clustering, a fundamental problem widely studied in the machine learning community and at venues like ICML.
- It introduces novel bounds for classical optimization problems and shows that they are better than the ones of the literature.
- The authors provide a rigorous theoretical analysis of their bounds, particularly those derived from the Multiplicative Weights Update (MWU) framework.
- A key insight is that existing heuristics, despite their worst-case bounds, perform exceptionally well on real-world instances, often producing near-optimal solutions. This finding is highly valuable and suggests the potential for broader applicability to other optimization problems.

Presentation.
- The paper is well written and a pleasure to read.


Weaknesses:

On originality:
- The proposed MWU-based algorithm for MinETCover builds upon a prior approach introduced in:
Fischer, N., Kipouridis, E., Klausen, J., & Thorup, M. (2025), A faster algorithm for constrained correlation clustering in STACS.
While the prior work updates the weight of a single triangle at each iteration, the current submission updates the weight of all triangles. The authors’ algorithm achieves better lower bounds and is faster, but it represents a small update of the existing method.
- The theoretical analysis of the algorithm, though new, follows standard techniques
- The other lower bound, GreedyPacking, is conceptually trivial. Surprisingly, however, it performs nearly as well as the MWU-based bound across a wide range of graphs. This suggests that even this simple approach could have sufficed to support the authors’ conclusions about instance-specific approximations.


Small comments:

On Gurobi and optimality.
Page 7, line 367: The statement "Gurobi... as it computes exact LP solutions" could be clarified. While Gurobi aims to compute exact solutions, it returns the best feasible solution found and its best lower bound when a time limit is imposed. This is evident in Figure 1, where Gurobi does not reach optimality for the largest instances.

Suggestion: Explicitly indicate in the table when the optimal solution is proven (e.g., by marking confirmed optimality).

Page 6: The phrase "To obtain solutions..., we use state-of-the-art heuristics... in Appendix B.1" would benefit from specificity. Since these heuristics are referenced later in the main text, it would be clearer to name them upfront (e.g., SCMLvo) to improve readability.

---

> ### Author Rebuttal · Authors · 2026-03-31
>
> We thank the reviewer for the positive evaluation of our paper.
>
> Regarding the similarities to Fischer et al., we stress that the two MWU approaches and their analysis differ conceptually. Specifically, our algorithm performs batched MWU updates, simultaneously updating the weights of many triangles per iteration. This leads to significant speedups in practice, but to analyze our algorithm we need to consider width-*dependent* oracles and build on the classical results by Plotkin et al. on solving packing--covering LPs with MWU. In contrast, the work of Fischer et al. relies on width-*independent* MWU updates, updating only one triangle per iteration. While most recent works on theoretically solving LPs with MWU are width-independent (**), it is not clear how to adapt their proofs to batch updates.  Therefore, while the algorithms might have similarities, the proofs are conceptually quite different. Theoretically, our paper is closer to the work by Plotkin et al. than to the approach by Fischer et al. In general, we agree with Reviewer k8Y3: "This is a nice application of MWU that needs a slight modification of the classic framework. The new ideas/proofs are not surprising but turn out to be quite useful for their application."
>
> Regarding the good performance of the simple GreedyPacking lower bound, we would like to stress that, even though it performs very well, its quality is not as good as the MWU bound we compute. While it is true that both lower bounds perform similarly for Correlation Clustering, for Max-Cut we observe better bounds from the MWU approach, improving the GreedyPacking bound by 5 percentage points on cond-mat and dblp, and by 1.5 percentage points on average. Furthermore, it is important to note that MWU is guaranteed to converge to a good solution, whereas GreedyPacking may have bigger gaps. Thus, in scenarios requiring strong worst-case guarantees, the MWU bound will be more reliable.
>
> We further thank the reviewer for pointing out improvement ideas in their small comments. We will address all of them in the final version of the paper.
>
> (**)
>
> Bhattacharya et al. (SODA'23) "Dynamic Algorithms for Packing-Covering LPs via Multiplicative Weight Updates"
>
> Checkuri et al. (SODA'20): "Fast LP-based Approximations for Geometric Packing and Covering Problems"

---

> > ### Author Rebuttal · Reviewer_Bkut · 2026-04-03
> >
> > I thank the authors for their response which addressed my questions.
> > I am considering increasing my score.

---

### Decision · Program_Chairs · 2026-04-30

**Decision:**

Accept (regular)

**Comment:**

The paper proposes an algorithm that computes lower bounds for several optimization problems: correlation clustering, max-cut, and likely (following from the discussion with reviewers) any sparse packing-covering LP. Such lower bounds are useful when the only algorithms we have for the problem are heuristics with no worst-case upper bound on approximation ratio, because with lower bounds one can at least estimate per instance empirical approximation ratio.

The algorithm is a somewhat natural generalization of previous work, but its analysis becomes much more technical, and empirical results are better.

The reviewers are unanimous that the paper should be accepted.